# The Hedgehog Pathway as a Therapeutic Target in Chronic Myeloid Leukemia

**DOI:** 10.3390/pharmaceutics15030958

**Published:** 2023-03-16

**Authors:** Andrew Wu, Kelly A. Turner, Adrian Woolfson, Xiaoyan Jiang

**Affiliations:** 1Terry Fox Laboratory, British Columbia Cancer Research Institute, Vancouver, BC V5Z 1L3, Canada; 2Interdisciplinary Oncology, Department of Medicine, University of British Columbia, Vancouver, BC V6T 1Z3, Canada; 3Department of Medical Genetics, University of British Columbia, Vancouver, BC V6T 1Z3, Canada; 4Replay Holdings Inc., 5555 Oberlin Drive, San Diego, CA 92121, USA

**Keywords:** leukemia, hematopoietic stem cells, leukemic stem cells, hedgehog, targeted therapy, AML, CML, small molecule inhibitors, combination therapy, signal transduction pathways

## Abstract

Despite the development of therapeutic agents that selectively target cancer cells, relapse driven by acquired drug resistance and resulting treatment failure remains a significant issue. The highly conserved Hedgehog (HH) signaling pathway performs multiple roles in both development and tissue homeostasis, and its aberrant regulation is known to drive the pathogenesis of numerous human malignancies. However, the role of HH signaling in mediating disease progression and drug resistance remains unclear. This is especially true for myeloid malignancies. The HH pathway, and in particular the protein Smoothened (SMO), has been shown to be essential for regulating stem cell fate in chronic myeloid leukemia (CML). Evidence suggests that HH pathway activity is critical for maintaining the drug-resistant properties and survival of CML leukemic stem cells (LSCs), and that dual inhibition of BCR-ABL1 and SMO may comprise an effective therapeutic strategy for the eradication of these cells in patients. This review will explore the evolutionary origins of HH signaling, highlighting its roles in development and disease, which are mediated by canonical and non-canonical HH signaling. Development of small molecule inhibitors of HH signaling and clinical trials using these inhibitors as therapeutic agents in cancer and their potential resistance mechanisms, are also discussed, with a focus on CML.

## 1. Introduction of the Hedgehog Signaling Pathway

The Hedgehog (HH) gene was first characterized in 1980 through a genetic screen of *Drosophila melanogaster* [1]. Larval mutants that were *hh*-null were characterized by a lawn of disorganized denticles on their ventral surface that resembled hedgehog spines [1]. The HH signaling pathway was subsequently found to be highly conserved across species. Its function turned out to be critical for both pattern formation in the developing embryo and adult homeostasis [2]. The HH signaling pathway is activated by one of three HH ligands found in mammals: Indian Hedgehog (IHH), Desert Hedgehog (DHH), and Sonic Hedgehog (SHH). Each has a distinct pattern of expression with a few overlapping functions [3,4]. IHH is involved in early hematopoiesis and skeletal development, whereas DHH plays a role in spermatogenesis [5,6,7,8]. SHH is the most well-studied HH homolog and is critical for the establishment of the left–right symmetry and development in vertebrates, ventral cell fates in the central nervous system, and antero-posterior limb development [9,10,11]. Although each protein regulates the formation of a different structure, their mechanisms of action are similar.

Once secreted, the ligands bind to Patched (PTCH), a 12-transmembrane protein, which inactivates its activity which in turn, constitutively inactivates a 7-transmembrane member of the G-protein-coupled receptor family called Smoothened (SMO) [12]. PTCH contains a sterol-sensing domain and has structural homology with members of the resistance-nodulation-division (RND) transporter family [13]. Mutations in the RND permease motif of PTCH diminish the repression of SMO, providing a possible mechanism of PTCH-mediated regulation of SMO involving the influx and efflux of regulatory molecules [14]. On the other hand, findings also suggest that PTCH induces the transportation of sterol-like ligands across the cell membrane to regulate SMO activity as several SMO agonists and antagonists have structural properties similar to sterols [15]. There are also several key intracellular mediators of the HH pathway downstream of SMO that operate in the primary cilium [16]. In response to HH pathway activation, SMO interacts with beta-arrestin and KIF3A and accumulates in the basal body of the primary cilium while the glioma-associated (GLI) zinc finger transcription factors, which are the main effectors of the pathway, complex with a repressor protein called Suppressor of Fused (SUFU) at the tip of the primary cilium [17,18,19,20,21].

In total, there are three GLI transcription factors, each of which performs a different role in modulating gene expression. While GLI1 is a transcriptional activator, GLI3 is a transcriptional repressor. GLI2, on the other hand, can act as either, depending on its post-transcriptional and post-translational modifications. Exposure to the HH ligand mediates dissociation of GLI-SUFU complexes so that activated GLI proteins can translocate to the nucleus where they promote the transcription of HH pathway target genes involved in proliferation and cell cycle progression [22,23]. Interestingly, the GLI transcription factors also appear to operate in a regulated feedback loop as other transcriptional target genes include PTCH and the GLI genes themselves. Without HH ligand binding, GLI remains bound to SUFU, eventually leading to phosphorylation and proteolytic processing of GLI to a repressive form (GLIR) which is a transcriptional inhibitor of HH pathway target genes (Figure 1A) [24]. These regulators of the HH signaling pathway are frequently compromised in malignancies, resulting in aberrant HH pathway activation and disease progression.

The HH signaling pathway may also be activated via non-canonical (SMO-independent) HH signaling mechanisms, which may also lead to cancer development. Non-canonical HH signaling involves the activation of GLI transcription factors by multiple different oncogenic signaling pathways, including: RAS/RAF/MEK/ERK, PI3K/AKT/mTOR, DYRK1B, IGF-1, and TGF-b [25,26]. GLI activity may also be modulated by oncogenes, tumor suppressors, and epigenetic modifiers. A comprehensive paper has recently been published that details the non-canonical HH signaling pathways, as well as their mechanisms of action and involvement in a variety of human malignancies [25].

## 2. Role of HH Signaling in Development

The HH pathway has been known to be most critical during development with HH homologues acting as morphogens, mitogens, or differentiation factors [27,28,29]. In the mouse embryo, SHH is expressed in the notochord at E9.5 which equates to a human embryo in approximately its third week of development [30]. Here, SHH acts as a morphogen and establishes a concentration gradient along the ventral neural tube to determine expression of transcription factors in spatially distinct progenitor domains to inform ventral CNS cell fates [31].

SHH acts as a mitogen during odontogenesis where it is produced and secreted by the dental epithelium in an autocrine manner via increased epithelial cell proliferation, and induces tooth germ growth by interacting with the underlying mesenchyme in a paracrine manner [32]. Additionally, SHH has also been shown to act as a differentiation factor in cell models of Parkinson’s disease [33]. Specifically, it was observed that SHH induced the differentiation of embryonic stem (ES) cells and induced pluripotent stem (iPS) cells to FOXA2^+^ neural progenitor cells (NPC), while other factors, such as FGF8 and retinoic acid, directed differentiation of NPCs to dopaminergic neurons [33].

## 3. HH Signaling in Hematopoiesis

Hematopoiesis is the process whereby cellular blood components are formed and replaced and starts during embryogenesis. In humans, after birth, this process is facilitated by multipotent, self-renewing hematopoietic stem cells (HSC) that reside in the bone marrow (BM), and can differentiate into myeloid- or lymphoid-restricted multipotent progenitors and subsequently commit to specific blood cell lineages [6,34,35]. HSCs are known to express HH pathway genes, such as PTCH1, SMO, and GLI, and stromal cells in the BM have been shown to have abundant SHH expression as well [36]. While HH signaling is essential for embryonic development, the role of HH signaling in normal hematopoiesis is still unclear. On one hand, studies have shown that human cord blood HSCs (Lin^−^CD34^+^CD38^−^) and adult murine HSCs (Lin^−^Sca1^+^cKit^+^, LSK) can be induced to proliferate continuously following HH pathway activation [36,37]. While this sustained proliferation eventually results in a depleted HSC pool, inhibition of SMO with the small molecule inhibitor cyclopamine was able to replenish the short-term self-renewal capacity of repopulating HSCs [38]. Other studies, on the other hand, have shown that inhibition of SMO with cyclopamine leads to deficient HSC function in primary and secondary transplant experiments in mice [38,39].

The role of downstream HH signaling molecules in regulating hematopoiesis has also been studied. In mice heterozygous for PTCH, the HH pathway activity was observed to be enhanced leading to observations of increased HSC progenitors particularly during short-term hematopoietic recovery following 5-fluorouracil treatment. Prolonged HH pathway activation in this model resulted in HSC exhaustion and limited self-renewal [37]. In GLI1 knockout mouse models, decreased HSC proliferation and enhanced HSC engraftment were observed together with impairment of myeloid differentiation [40,41]. Normal mice also appeared to have more long-term quiescent HSCs compared to mice with SMO knockdown [41]. Despite these experimental findings which detail important roles of HH signaling in hematopoiesis, it has been suggested that HH signaling is dispensable. Other studies have shown that SMO inhibition via inducible Cre knockout systems in mouse models does not have a significant impact on HSC homeostasis, and HSC gene expression analysis reveals few changes in the absence or presence of SMO [39,42].

## 4. HH Signaling in Cancer and Development of Small Molecule SMO Inhibitors

As a critical regulator of cell growth and differentiation, it is unsurprising that mutations in different HH pathway members may lead to pathway hyperactivation that can predispose to cancer. Aberrant HH signaling activation in cancer is classified as either type I (ligand-independent) signaling, type II (ligand-dependent) autocrine/juxtacrine signaling, or type III (ligand-dependent) paracrine signaling (Figure 1) [43,44]. Type I HH pathway activation includes gain-of-function mutations in SHH that contribute to the formation of basal cell carcinoma, and heterozygous loss-of-function mutations in PTCH1 which cause Gorlin syndrome, basal cell cancers, and a range of other neoplasms including medulloblastoma [44,45,46]. Other malignancies such as colorectal, breast, prostate, liver, small cell lung and brain cancers have been shown to be characterized by type II ligand-dependent autocrine/juxtacrine signaling, in which the HH ligand is actively secreted and taken up by tumor cells (Figure 1C) [44,47,48]. Type III (ligand-dependent) paracrine signaling involves the secretion of HH ligands from cancer cells that activate downstream HH signaling in stromal cells via PTCH1, and is most frequently seen in prostate, pancreatic, and colon cancers (Figure 1B) [44,49]. This results in the release of growth factors by stromal cells, which further drive tumorigenesis. Reverse paracrine signaling may also occur, principally in hematologic malignancies such as acute myeloid leukemia (AML) and CML, and involves the release of ligands from stromal cells that enhance the growth and survival of malignant cells (Figure 1D) [38,50,51].

Numerous small molecule inhibitors of HH signaling have been developed for the treatment of a variety of human cancers. Many of these are direct SMO antagonists, binding to the SMO receptor and blocking downstream activation of GLI and other HH effector proteins [26,52]. These include vismodegib, the first FDA-approved SMO inhibitor, licensed for the treatment of basal cell carcinoma (BCC), and LY2940680 (taladegib) which is being developed for the treatment of BCC, small cell lung carcinoma, and a range of other solid tumors, and is currently undergoing phase 2 investigational studies (Figure 2). Established SMO antagonists that have demonstrated efficacy in targeting myeloid leukemias include: NVP-LDE225 (sonidegib/erismodegib), PF-04449913 (glasdegib), IPI-926, and BMS-933923 (Figure 2) [26,53].

## 5. HH Signaling in CML

CML is a myeloproliferative disease characterized by the presence of a constitutively active BCR-ABL1 tyrosine kinase fusion gene that plays a key role in driving the oncogenic signaling pathways of the disease. While the advent of tyrosine kinase inhibitors (TKI) has been effective in managing the disease, functional cure is challenging to attain due to BCR-ABL1-independent resistance mechanisms and various treatment escape mechanisms available to leukemic stem cells (LSC) [54,55,56,57]. The HH pathway is one such BCR-ABL1-independent mechanism promoting TKI resistance, which was not appreciated until two landmark studies demonstrated that intact HH signaling is required for LSC maintenance [38,51].

Initial studies revealed that the expression of GLI1 and PTCH1 progressively increased in CML patient cells from the chronic phase (CP) through the accelerated phase (AP) and blast crisis phase (BC) of the disease [38,58,59]. Additionally, it was observed that murine fetal SMO^−/−^ HSCs retrovirally transduced with BCR-ABL1/GFP were less able to induce leukemia when injected into immunodeficient mice; disease latency was extended by more than three months in this group compared to the control group, and only 60% of recipients developed lethal disease [38]. Furthermore, BM from the diseased mice was retransplanted into secondary hosts and no mice receiving SMO^−/−^/BCR-ABL1/GFP^+^ cells developed leukemia, unlike controls, who developed leukemia within two months [38]. It was also observed that a pharmacological suppression of SMO via cyclopamine was able to increase survivability of leukemic mice with significantly less BCR-ABL1/GFP^+^ LSK cells in their BM (1% vs. 14%, *p* < 0.05) compared to controls [38,51]. A combination of cyclopamine with the second generation TKI Nilotinib in vitro and in vivo was also found to be superior to TKI monotherapy at reducing the colony-forming ability of patient-derived CML stem and progenitor cells and decreased spleen and liver weights in mice [38]. Overall, these data and related reports suggested that intact HH signaling, mediated by SMO, was essential for the expansion of LSCs both in vitro and in vivo.

In order to further elucidate the significance of the HH pathway in CML, RNA sequencing analysis was performed on CD34^+^ stem/progenitor cells obtained at diagnosis from six CP-CML patients and three healthy bone marrow (HBM) controls [60,61]. Among 27 differentially expressed HH pathway genes found, SMO and GLI2 in particular, were highly upregulated in Imatinib (IM) non-responders compared with responders [60,61]. These two genes were also observed to be differentially expressed between primitive and mature subpopulations where they were more highly expressed in the stem-enriched subpopulation (Lin^−^CD34^+^38^−^) compared to both progenitors (Lin^−^CD34^+^38^+^) and the more mature (CD34^−^) subpopulations [60,61]. To follow up, CD34^+^ cells from IM responders and IM non-responders were treated with the SMO inhibitor PF-04449913 and it was found that IM non-responders were more sensitive to SMO inhibition compared with IM responders, with respect to cell survivability, replating potential, and colony-forming ability following long-term (>6-weeks) culture [60,61]. A dual treatment strategy comprising the second generation TKI Bosutinib in combination with PF-04449913 in CD34^+^ IM non-responder cells showed significant improvements in reducing colony-forming ability and replating potential compared with either agent alone [60,61]. This suggested that HH activity is required for the maintenance of LSCs and that dual inhibition of the BCR-ABL1 and HH pathway, and especially SMO, may provide a compelling strategy for targeting drug-insensitive LSCs.

The effects of PF-04449913 in CML were further explored using RNA-seq analysis of seven CP-CML, six BC-CML, three healthy cord blood, and three healthy PB progenitor (Lin^−^CD34^+^CD38^+^) samples. GLI2 was not only found to be significantly differentially expressed between CP-CML and healthy controls, but even more highly expressed in BC-CML samples by up to 7-fold compared with CP-CML samples [60,61]. Flow cytometry-sorted BC-CML progenitor cells also demonstrated reduced survival in response to a seven day treatment of PF-04449913 compared to normal cord blood cells in a coculture experiment with SL/M2 mouse stromal cells modified to produce human growth factors [60,61]. The findings in this experiment suggest that BC-CML cells are selectively targeted by PF-04449913 as a result of likely GLI2 dependency [60,61]. Interestingly, it was also found that GLI2 may also play a role in regulating CML LSC dormancy. CP-CML progenitor cells transduced with GLI2 were found to preferentially reside in the G0 phase of the cell cycle, compared with cells transduced with the empty vector control, or a GLI2 deletion mutant [60,61]. Thus, it is possible that the HH-mediated regulation of LSC dormancy may be a contributing factor towards resistance to therapy [60,61]. The combination of PF-04449913 with another second generation TKI, Dasatinib, has also been shown to selectively inhibit engraftment of primary Lin^−^CD34^+^CD38^+^ CML LSCs in Rag2^−/−^g_c_^−/−^ mice and to eradicate the formation of myeloid sarcomas [62].

In another study, a combination of the second generation TKI Nilotinib with the SMO inhibitor LDE225 was found to be effective in inhibiting CP-CML cells [59]. The expression of HH pathway-associated genes was investigated in the HSC, common myeloid progenitor (CMP), granulocyte-monocyte progenitor (GMP), and megakaryocyte-erythrocyte progenitor (MEP) subpopulations in human BM and CP-CML samples. Colony forming cell (CFC) replating and long-term culture initiating cell (LTC-IC) assays with the Nilotinib and LDE225 combination also showed significant reductions in CFC output in replating assays compared to the untreated control [59]. However, no significant reduction in LTC-IC-derived CFC output was observed with LDE225 alone, or in combination with Nilotinib. Significantly reducing CFC output was, furthermore, unable to eradicate leukemic cells completely [59]. This lack of effect with SMO inhibitor in both studies suggested that the HH pathway may be activated in a non-canonical manner [25,26].

A recent study using CML cell lines and IM-resistant patient samples found that autocrine HH signaling may also promote resistance by upregulating BCL-2 (Figure 2) [63]. In this study, SHH, PTCH1, and GLI1 expression levels were found to be upregulated in IM-resistant patient samples as well as in CML variants relative to CP-CML patients and the K562 cell line [63]. Through the analysis of LSC markers and survival assays in SHH-overexpressed cells, the study confirmed prior reports that HH signaling imparts stemness and survival advantage with or without IM treatment selection pressure [63]. Aside from increased levels of the HH pathway proteins in CML variants and IM-resistant samples, the study also demonstrated that these samples had upregulated BCL-2 levels and that inhibiting either SHH or BCL-2 could resensitize leukemic cells to IM [63]. Interestingly, it was also determined that exovesicular SHH obtained from IM-resistant BM plasma facilitated IM resistance in K562 cells, as opposed to free non-vesicular SHH [63]. Thus, this study suggests that exovesicular SHH and BCL-2 in CML patients may have predictive clinical utility and comprise druggable targets for combination therapy to target IM resistance [63]. For CML variants with high levels of SHH or BCL-2, the application of Venetoclax could be a relevant therapeutic option as it is a highly selective BCL-2 inhibitor that also has SMO-inhibiting properties (Figure 2) [64,65,66]. Despite these promising results that provide further evidence for the relevance of SHH signaling in CML, it is notable that most of the experiments were performed on the K562 cell line. It is consequently imperative to further investigate whether these results can be generalized to primary patient samples.

The HH signaling pathway in CML has been shown to undergo crosstalk with other coactivated oncogenic pathways, including the Wnt/β-catenin signaling pathway, as observed in CD34^+^ and c-kit^+^ CML progenitor cells [58]. This was explored in K562 cells treated with the SHH blocking antibody 5E1, which demonstrated associated downregulation of C-MYC, an established target of β-catenin, and the apoptosis antagonist BCL-2 (Figure 2) [58]. P21, a cell cycle checkpoint inhibitor, was also found to be upregulated, inducing G2/M cell cycle arrest (Figure 2) [58]. These observations suggested that SHH signaling may determine cell fate by mediating apoptosis and cell cycle arrest of CML cells via the β-catenin pathway (Figure 2). Other pathways that integrated with SHH signaling in other cancers, include the extracellular regulated kinase (ERK), protein kinase C-δ (PKCδ), transforming growth factor β (TGFβ), and the mitogen-activated protein/extracellular signal-regulated kinase (MEK) pathways [67,68]. These pathways have been shown to behave as SMO-independent enhancers of GLI1/2 expression in numerous solid cancers and confer resistance to front-line SMO inhibition therapies. Thus, the development of strategies to treat CML variants characterized by non-canonical HH signaling may require patient prescreening for resistance to SMO inhibition as well as to combination treatments targeting multiple pathways.

## 6. Clinical Implications

The clinical effectiveness of a combination therapy comprising a TKI and SMO inhibitor combination as an LSC-targeted therapy in CML has yet to be fully determined. In a phase 1 safety and pharmacokinetics study of PF-04449913 in myeloid malignancies, five CML patients were enrolled (two CP-CML and three BC-CML) [69]. One patient with BC-CML achieved a partial cytogenetic response (pCyR) [69], but there were no other responders. Sixty percent of the treated patients experienced non-hematological adverse events, the most common being dysgeusia (28%), decreased appetite (19%), and alopecia (15%). All were grade 1–3 in severity [69]. While the authors concluded that a phase 2 study was not warranted for CML, they recommended that efficacy-focused studies with combination therapies be performed. The efficacy of an additional oral HH inhibitor, IPI-926, was also assessed in 14 myelofibrosis patients in a phase 2 clinical study [70]. Patients showed either no response, or a minimal response and the study was discontinued.

Encouraging results for HH inhibitors in AML and myelodysplastic syndrome (MDS) have been reported in several clinical trials [71]. A phase 2 study of PF-04449913 in combination with the chemotherapeutic agent Ara-C showed that patients in the combination arm had a longer median overall survival (mOS) compared with patients treated with Ara-C monotherapy [72]. Patients stratified as having an intermediate cytogenetic risk with the combination exhibited a mOS of 12.2 months vs. 6 months for Ara-C alone, while low risk patients on the combination had a mOS of 4.4 vs. 2.3 months when treated with Ara-C alone [72]. Adverse events associated with PF-04449913 treatment were similar to those described in the phase 1 study, and overall had an acceptable safety profile [69]. Venetoclax has already been approved by the FDA for use in newly diagnosed AML in combination with a hypomethylating agent or low dose cytarabine [65,73,74]. Although these regimens primarily yield anti-leukemic effects via BCL-2 inhibition, the SMO-inhibiting properties of venetoclax may demonstrate further benefits, especially in AML variants with amplified HH signaling [66].

Two other clinical trials on dual TKI and SMO inhibitor treatments have also been conducted for CML with one assessing the combination of Dasatinib with BMS-933923 and the other evaluating Nilotinib with LDE225. Unfortunately, both trials reported only moderate responses for these combination therapies while also reporting poor tolerability and adverse side effects in patients, eventually leading to these trials to be discontinued [75]. Thus, despite the efficacy of SMO inhibitors alone or in combination with TKIs in preclinical settings, efforts to implement these strategies for CML patient treatment have largely been hindered due to toxicity associated with SMO inhibition [76]. While HH signaling is mostly critical during development, adverse effects characteristic of SMO inhibition such as alopecia, muscle spasms, and dysgeusia arise because HH signaling in normal adult tissues is required for repair, maintenance, and stem cell proliferation in hair, muscle, and taste bud cells which may be inadvertently affected by SMO-inhibitor treatment [77]. Therefore, it is critical to continue identifying novel therapeutic agents and strategies to overcome the issue of poor SMO-inhibitor tolerability.

It has been reported more recently that sulforaphane (SFN), a compound found in cruciferous vegetables, can induce apoptosis in leukemia cells and regulate the proliferation of LSCs in vitro and in vivo via HH signaling pathway inhibition [78]. SFN treatment in KG1a and KG1 leukemic cells reduced transcript levels of HH pathway proteins such as SMO, PTCH1, and GLI1 [78]. Additionally, SFN was able to suppress proliferation and colony formation in SHH-overexpressing KG1a/KG1 cells [78]. However, in KG1a/KG1 cells with SHH knockdown, SFN appeared to have minimal effects. This suggests that the anti-proliferative effects of SFN are mediated by the HH signaling pathway [78]. These results are further supported by an in vivo xenograft model where NOD/SCID mice were injected with CD34^+^KG1a cells and treated with SFN. Tumor volume and tumor weight were both reduced compared to the control group in conjunction with reduced cell proliferation and IHC signal expression of SMO, PTCH1, and GLI1 [78]. It is, however, yet to be determined whether these discoveries can be repeated in human primary patient cells.

## 7. Mechanisms of Resistance to Small Molecule Inhibitors of HH Signaling

Some cancers are able to develop mechanisms of acquired resistance to the traditional SMO antagonists detailed above. Since the isolation of the D473H SMO mutant, which is an aspartic acid to histidine point mutation that prevents inhibitor binding, numerous other SMO point mutations have also been discovered [52,79]. Aside from preventing ligand binding, these mutations may also induce SMO reactivation as well as additional mutations that may lead to the upregulation of GLI and other HH target genes, all of which confer SMO-inhibitor resistance [52,79].

To overcome these SMO-dependent resistance mechanisms, second generation SMO inhibitors have been discovered through high throughput screening. These include Compound 5, which showed tumor-reducing effects in vismodegib-resistant in vivo tumor models (Figure 2) [26]. Other inhibitors such as 0025A, HH-1/13/20, ZINC12368305, and LEQ-506 bind SMO with a greater potency and inhibit downstream HH signaling irrespective of the D473H mutation [26,80]. Other inhibitors, such as venetoclax, target the cysteine-rich domain of the SMO protein and can also overcome the D473H mutation [26]. As the chemical properties and mechanism of action of these inhibitors and others have already been reviewed, they will not be discussed further.

As discussed above, non-canonical HH signaling or SMO-independent mechanisms are also drivers of resistance to SMO inhibitors. For example, in gastrointestinal cancer, HH signaling increases gene expression of ATP-binding cassette subfamily G member 2 (ABCG2) via GL1, enhancing drug efflux and reducing drug concentrations in cancer cells, which could generate chemoresistance [81]. In pancreatic and colorectal cancers, HH signaling promotes cancer cell stemness through elevated expressions of hypoxia inducible factor 1a (HIF1a) and the PI3K/AKT signaling pathway [25,82,83]. The RAS-RAF-MEK-ERK pathways have also been shown to enhance nuclear localization and transcriptional activity of GLI1 to circumvent SMO inhibition in melanoma cells [84]. Additionally, the HH pathway has been shown to increase drug resistance in pancreatic ductal adenocarcinoma by modulating gene expression in the tumor microenvironment [85]. HH pathway-modulated drug efflux and drug metabolism are also contributors to drug resistance in AML, via chromosomal amplification of GLI2 and associated upregulation of P-glycoprotein and UDP glucuronosyltransferase (UGT1A) [71,86,87]. Other noncanonical mechanisms of resistance to SMO inhibitors, such as the effects of protein kinases and chromatin modulators, have also been described [52].

An important question raised by these studies of resistance to SMO inhibitors relates to the origin of HH pathway activation in CML. Canonical HH signaling commences with the binding of HH ligands to the transmembrane protein PTCH. HH ligands can be supplied to target cells in an autocrine manner, or from surrounding stromal cells or CML cells in a paracrine manner (Figure 1B–D). However, the Sadarangani study assessing the effects of PF-04449913 in CML with RNA-seq analysis did not report the extent of SHH, IHH, or DHH expression, so it is not known whether the HH signaling is being activated in a canonical or non-canonical manner in CML cells [61]. Characterization of the origin and type of HH signaling in CML is important, as it specifies which cells will be sensitive to treatment. One way to determine whether stromal cells provide HH ligands is to analyze the expression of SHH, IHH, and DHH at the gene and protein levels using qRT-PCR and Western blotting, respectively, in normal BM stromal cells. To determine whether HH signaling is being activated in a non-canonical fashion, a GLI inhibitor could also be used, as GLI is a direct downstream effector of SMO. Although there are currently no GLI inhibitors in clinical trials for CML, GANT61 is a dual GLI1/2 inhibitor that has demonstrated effectiveness in multiple cancer models [88,89,90,91,92]. Other GLI inhibiting compounds such as BET inhibitors and arsenic trioxide (ATO) have also shown some promise in the clinic for some cancers, including acute promyelocytic leukemia [80,93]. Additionally, GLI expression can be regulated by histone deacetylase (HDAC) inhibitors which are already approved for hematologic malignancies [80]. Thus, it will be important to determine whether CML cells are more sensitive to GLI inhibition compared to SMO inhibition, as this would suggest that the HH pathway is activated in a SMO-independent manner and would comprise an alternative route for overcoming resistance to SMO inhibitors.

## 8. Conclusions and Future Directions

Since the characterization of the HH gene in 1980, significant advancements have been made in establishing its importance both in the development and in driving the pathogenesis of cancer. Despite the development of several HH pathway inhibitors that have been tested in clinical studies for a range of different solid tumor pathologies, clinical studies of these inhibitors in hematological malignancies, especially CML, remain limited. Even for existing SMO inhibitors, toxicity and resistance remain significant issues. Thus, complementary therapeutic strategies including the use of next-generation SMO inhibitors, combination therapies, and genetic suppression of HH pathway proteins using shRNA or CRISPR-Cas9, are urgently required.

## Figures and Tables

**Figure 1 pharmaceutics-15-00958-f001:**
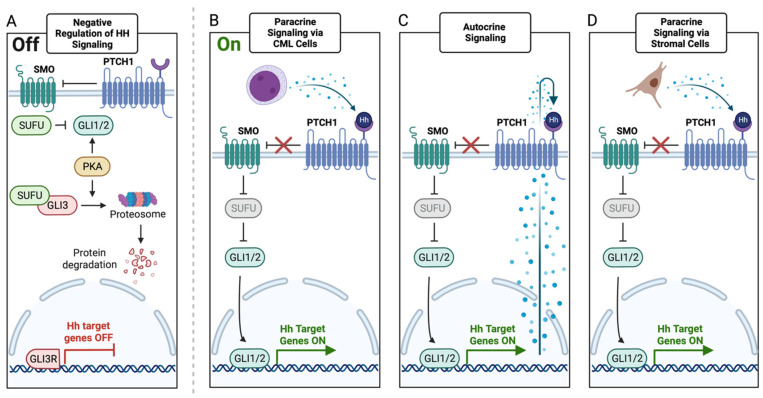
Canonical Mechanisms of Hedgehog Signaling. (**A**) In the absence of HH ligands, PTCH1 inhibits SMO and promotes SUFU-mediated inhibition of GLI1/2 and GLI3 proteolytic degradation to GLIR which translocates to the nucleus as a transcriptional repressor. (**B**) Paracrine, or type III signaling, involves the binding of the HH ligand, which is secreted by cancer cells to the PTCH1 receptor which prevents PTCH-mediated inhibition of SMO, lifting the SUFU-mediated repression of GLI1/2 and promoting the transcription of HH target genes. (**C**) Type II ligand-dependent autocrine/juxtacrine signaling involves the secretion and exportation of HH ligands from the cell which then bind to the cell’s own PTCH1 receptor to trigger the cascade seen in type I signaling. (**D**) Reverse paracrine signaling involves the secretion of HH ligands from stromal cells in the tumor microenvironment and is the HH activation mechanism most observed in myeloid malignancies.

**Figure 2 pharmaceutics-15-00958-f002:**
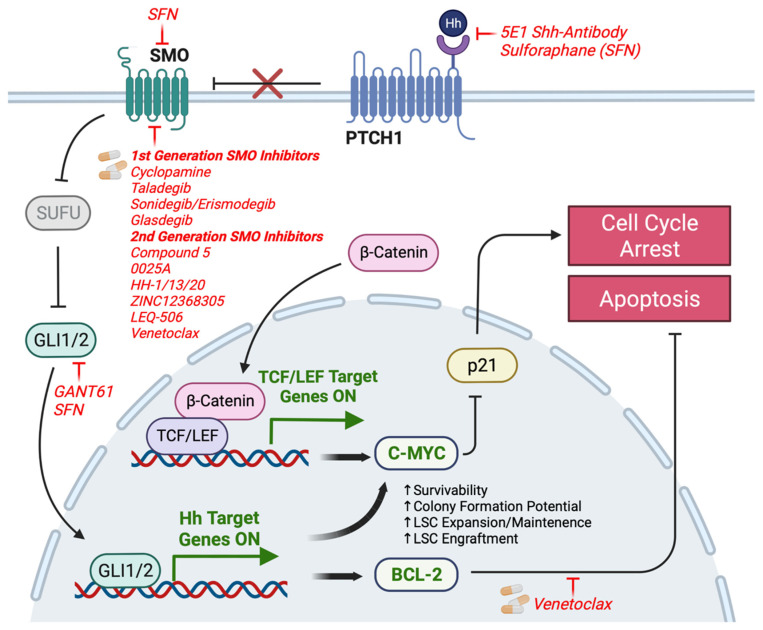
Hedgehog Signaling in CML. The HH signaling pathway has been implicated in CML as a BCR-ABL1-independent pathway which contributes to TKI resistance and disease progression. In a CML-specific context, GLI1/2-mediated transcription upregulates the anti-apoptotic gene BCL-2 as well as the oncogene C-MYC which promotes LSC survivability and proliferation. SMO-independent or non-canonical HH pathways, such as the Wnt/β-Catenin pathway, also contribute to C-MYC upregulation. To overcome SMO-inhibitor resistance, various strategies have been investigated, such as the use of next-generation SMO inhibitors, anti-SHH antibodies, and GLI inhibitors.

## Data Availability

No new data were generated for this review. All results described are cited and data sharing is not applicable to this article.

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
