# Peer review of "The Hedgehog Pathway as a Therapeutic Target in Chronic Myeloid Leukemia"

_pharmaceutics, 2023, doi:10.3390/pharmaceutics15030958_

Round 1

Reviewer 1 Report

the paper is good and well written

Author Response

Reviewer #1

We thank the reviewer for your positive review of our paper.

Reviewer #1

We thank the reviewer for your positive review of our paper.

Reviewer #1

We thank the reviewer for your positive review of our paper.

Reviewer #1

We thank the reviewer for your positive review of our paper.

Reviewer #1

We thank the reviewer for your positive review of our paper.

Reviewer 2 Report

The review «  The Hedgehog Pathway as a Therapeutic Target in Chronic 2 Myeloid Leukemia” from Wu and co-author is very well documented and written

I have no major comment

Minor comments

1.      At least to clinical studies are registered on clinical trial.gov in the field of CML. These studies may be mentionned. 

-         NCT01218477 testing the combination of BMS-833923 and dasatinib

-         NCT01456676 testing the combination of LDE225 and nilotinib

2.      Despite efficacy in the preclinical setting all strategies using SMO inhibitors in CML patients failed : the authors may discuss in more detail the problem of irreversible toxicities

Author Response

Please see a attached letter

Reviewer 3 Report

I found the manuscript “The Hedgehog Pathway as a Therapeutic Target in Chronic 2 Myeloid Leukemia” interesting and well writing. I thinks to the readers of the journal will be pleased. I just have some suggestions:

Avoid the use of Reviews, change by experimental papers

Verify the terms pluripotent and multipotent

The section of “3. HH Signaling in Hematopoiesis”, could be improved describing more experimental results 

Author Response

Please find the attached letter

Author Response

Please see the attached letter
